# Trait self-control does not predict attentional control: Evidence from a novel attention capture paradigm

**Michael A. Dieciuc\*, Heather M. Maranges, Walter R. Boot**

Florida State University Department of Psychology, Tallahassee, Florida, United States of America

\* dieciuc@psy.fsu.edu

## Abstract

To what extent are low-level visual and attentional phenomena related to higher-level personality traits? Trait self-control is thought to modulate behavior via two separate mechanisms: 1) by preventing initial temptation and, 2) by inhibiting temptation when it occurs (disengagement). Similarly, the control of visual attention often entails preventing initial distraction by irrelevant but tempting (goal-similar) objects, and disengaging attention when it has been inappropriately captured. Given these similarities, we examined whether individuals higher versus lower in trait self-control would differ in their susceptibility to attention capture using mouse-tracking as a sensitive, online measure of how attentional dynamics resolve over time and space in response to a distracting visual cue. Using a variety of metrics of attention capture, we found that differences among people in trait self-control did not predict initial selection of visual information nor subsequent disengagement. Overall, these results suggest that trait self-control and attention capture operate via separate mechanisms.

**Data Availability Statement:** Please note that our hypotheses, predictions, and analyses were preregistered (https://osf.io/twhnj/) and our raw data, scripts, and Supplemental Materials are publicly available (https://osf.io/xnc7k/).

## Introduction

Does the ability to successfully ignore colorful but task-irrelevant billboards while driving have anything in common with the ability to successfully refuse a delicious but unhealthy slice of cake? Both tasks involve inhibitory control—one to suppress distraction, the other to suppress temptation. Are these mechanisms related to one another or are they orthogonal? One possibility is that the cognitive mechanisms involved in demonstrating the self-control to say "no" to an unnecessary slice of cake when one holds the goal of eating healthfully are similar to the mechanisms involved in resisting visual distraction. Another possibility is that the macro-level processes involved in self-control are orthogonal to micro-level processes involved in attention capture. Here, we explore whether differences in trait self-control influence attention capture.

### What is attention capture?

Attention capture is the involuntary and transient prioritization of task-irrelevant stimuli. In other words, it is when we temporarily lose control of our attention and attend to something

**Funding:** The author(s) received no specific funding for this work.

**Competing interests:** The authors have declared that no competing interests exist.

irrelevant to the task at hand. One of the most popular methods for examining this phenomenon is the contingent capture paradigm [1–2]. In a sense, this paradigm can be thought to measure distraction by irrelevant but tempting (goal-similar) objects. Participants are instructed to search for and identify a specific target (e.g., a red symbol among symbols of other colors), and response times are recorded. Prior to searching for the target, the display screen is cued with an irrelevant distractor that either shares the defining feature of the target (e.g., a red flash of color) or does not match the target (e.g., a green flash of color or the abrupt appearance of a new object). The typical finding is that only distractors that match what people are looking for capture attention (slowing responses when they appear at non-target locations), whereas ones that do not match have little ability to do so. For example, an observer searching for a red X may involuntarily direct attention toward an irrelevant flash of red, but not an irrelevant flash of green, even though both represent salient environmental change. Some researchers suggest that this pattern of attention capture is primarily related to the mechanism of attentional disengagement [3–4]. They argue that all salient stimuli are capable of capturing attention, but disengagement from that distractor is delayed to the extent that the distractor shares features with the search target, producing longer response times.

## Individual differences in capture

Notably, susceptibility to attention capture differs across people: some are more or less prone to having their attention captured. People high in working memory capacity exhibit less capture by irrelevant information [5–6]. Similarly, higher working memory capacity is associated with faster recovery from capture [7]. In addition to the working memory literature, some work suggests that action video game experience reduces the involuntary capture of attention [8]. Finally, there is other work suggesting that mood may influence attention capture; specifically, it has been shown that depression can increase or decrease capture effects depending on the nature of the distractor [9]. Collectively, these studies suggest that basic mechanisms of attention may be susceptible to individual difference factors—with some factors increasing susceptibility to capture and others decreasing it. Along these same lines, we ask if trait self-control also influences susceptibility to attention capture.

## Self-control

Self-control can be conceptualized as the ability to (a) override or inhibit prepotent responses, including thoughts, emotions, and behaviors, and (b) replace them with responses more consistent with social norms or one's long-term goals [10–11]. Self-control varies according to situational demands or constraints [12] and also among individuals [13]. This stable trait is the focus of the current work. High trait self-control facilitates waiting for larger, later rewards over smaller, sooner rewards [14] and is associated with various positive life outcomes, including academic and career success, quality of interpersonal relationships, psychological wellbeing, avoidance of substance abuse and crime, and relatively better health [13, 15, 16].

How does self-control work and what mechanisms does it recruit? Self-control is thought to work by employing executive functions toward maintaining goal-pursuit [17–18]. Researchers have argued that self-control depends on behavioral inhibition, task-switching, and working memory [17–18]. It is also associated with deliberative cognitive processing, such as planning, decision making, and impulse control [15]; consideration of future consequences [13]; and need for cognition [19].

In the past, researchers assumed that people high in trait self-control differed from those low in trait self-control merely in the extent to which they could *overcome* temptation to violate a norm or undermine goals [20]. Augmenting that view, a burgeoning body of literature

suggests that people high in trait self-control may not experience temptation or distracting desires to the same extent that people low in this trait do [17] and try to avoid or preclude distractions [21] in addition to resisting temptations more effectively [17]. Researchers have used experience sampling methods to map out everyday self-regulatory processes and found that people higher (vs. lower) in trait self-control did not experience as many urges or desires that conflicted with more important goals [22]. Of the temptations that broke through, people higher in trait self-control felt less tension and more often succeeded in overcoming them.

Moreover, people high (vs. low) in trait self-control actively avoid distractions in their environment that might undermine performance on an important task and when speed or accuracy garner larger rewards. For example, in laboratory settings, people high but not low in self-control chose in advance to forgo breaks in an economics experiment to read entertaining stories [23], to wait for a distraction-free room to become available to optimize performance on an anagram task rather than work in a noisy room that was available immediately [21], and to view a boring standard black and white version of an anagram task instead of a more aesthetically pleasing but distracting one (i.e., included pictures of classic and modern artwork on either side; [21]). Other work suggests that people higher (vs. lower) in trait self-control avoid goal conflicts by automatizing particular behaviors (i.e., forming habits), including in health and academic domains [12, 24, 25].

In addition to these macro-level behavioral differences, self-control is also associated with low-level cognitive processes, such as working memory [17–18]. Given that differences in working memory are associated with differences in attention capture [5, 7], it is reasonable to suspect that differences in trait self-control may be associated with differences in attention capture. Specifically, we might expect that people higher in trait self-control are less susceptible to attention capture and are better at filtering out goal-irrelevant stimuli. This would fit with the idea that effective self-control entails preclusion of distraction. On the other hand, it may be the case that people high (vs. low) in trait self-control cannot help but initially attend to distracting stimuli in the perceptual field, but they can better disengage from those stimuli. This would be consistent with the traditional view that self-control entails overcoming that which does not facilitate one's goals [20]. Given that people with high self-control experience fewer distractions, preclude more distractions, and more easily overcome distractions relative to those low in trait self-control [17], it may even be the case that trait self-control is positively associated with both the initial selection of visual information and the subsequent disengagement from irrelevant visual information.

## Current study

### Research question

Self-control involves mechanisms of suppression and inhibition. Attention capture also involves analogous mechanisms of suppression and disengagement. To what extent are these mechanisms that operate on very different levels—self-control on a macro-level and attention capture on a micro-level—related to one another? On the one hand, we might expect that differences in self-control could influence basic visual attention processes. If so, the question then becomes: which aspects of attention capture does it affect—selection, disengagement, or both? On the other hand, it has also been argued that attention capture has strong bottom-up components [3]. The strong version of this approach is that bottom-up processes are impenetrable and unaffected by top-down processes. In this case, we would expect differences in trait self-control to be unrelated to differences in attention capture, but may be more related to disengagement.

## Mouse-tracking

In order to test whether self-control is associated with attention capture—and if so, which mechanisms it affects—we employed a computer mouse-tracking paradigm [26–29]. Participants selected a colored X that appeared on the screen while the x and y the coordinates of their responses were continuously recorded. The underlying assumption of mouse-tracking is that the partial and tentative conclusions of perceptuo-cognitive systems continuously feed into and affect the motor system [28, 30, 31]. Thus, the parallel influence of various cognitive processes can be seen in the unfolding trajectory across space and time. The advantage of this methodology is that it provides a high-resolution measure of online behavior, one capable of disentangling mechanisms of selection and disengagement from one another. Thus, the assumption is that if a participant's attention is captured, it will be reflected in the trajectory of their mouse movement [32–35].

Our decision to use mouse-tracking instead of a similar methodology like eye-tracking was based on a number of advantages related to measuring how cognitive processes play out over time. While saccades are largely ballistic in nature and their flight paths are often completed in tens of milliseconds, mouse-tracking unfolds over a much larger timescale, providing a rich look at both spatial and temporal dimensions of behavior. On another level, mouse-tracking was chosen over eye-tracking for a number of practical reasons. Mouse-tracking software is freely available [27, 29] to anyone who has access to a computer. This high-accessibility makes it an appealing method of conducting research, one that is particularly conducive to encouraging replication and reproducibility. Finally, mouse-tracking is a relatively new methodology compared to eye-tracking, especially within the field of attention capture and self-control; thus, given the novelty of the methodology, we wanted to deepen the field's knowledge.

## Predictions

Broadly speaking, there are four different ways trait self-control and attention capture can interrelate. One, self-control may only correlate with selectivity—the degree to which features similar to our goals (i.e., matching cues) capture attention over and above features that are dissimilar with our goals (i.e., mismatching cues). People high in self-control may be more resistant to the involuntary capture of their attention by visually salient but goal-similar features. Two, self-control may only correlate with disengagement—the ability to release attention after it has been captured. People high in self-control may be just as susceptible to being captured by salient stimuli but may be better at disengaging from it. Three, self-control may correlate with both selectivity and disengagement. People high in self-control may be less susceptible to the involuntary capture of attention by a goal-similar distractor and better at disengaging from it when they are captured. Finally, self-control may be uncorrelated to either selectivity or disengagement. There may be no difference between people with high and low self-control in how their attention is captured.

To preview our results, our data are most consistent with the fourth possibility. Namely, self-control scores did not correlate with either selectivity or disengagement. This suggests that attention capture and trait self-control operate via different mechanisms.

## Methods

All procedures complied with and were approved by Florida State University's IRB; participants were compensated with course credit. Data were collected from 103 participants; however, three participants were excluded from analysis because they did not complete the self-control survey (final $N = 100$, 71 females, $M_{age} = 20.03$). This sample size was determined by

an a prior power analysis. Using an alpha level of .05 and a power level of .8, our sample of 100 participants is sufficiently powered to detect a medium correlation ($r$ = .28; see [36]).

Note, the procedures, apparatus, and preprocessing in this paper were identical to previous research [35]. Our hypotheses, predictions, and analyses were preregistered (https://osf.io/twhnj/) and our raw data, scripts, and Supplemental Materials are publicly available (https://osf.io/xnc7k/).

## Apparatus and stimuli

The experiment was programmed, preprocessed, and analyzed using the following software: Programming was done in OpenSesame version 3.1.9 [37] using the mousetrap package [29] and the legacy backend; Data importing and preprocessing were done in R [38] using the *readbulk* [39] and *mousetrap* [40] packages; Data were analyzed using a combination of R packages [41] and *JASP* 0.8.2.0 [42]. Data were collected on a computer running Windows 7 with default mouse settings. In OpenSesame, the script was set to record coordinates at a temporal resolution of 10 ms. Participants sat approximately 30 cm from an 18-inch Dell Trinitron CRT monitor. The resolution of the screen was set to 1280 X 800 by OpenSesame.

Participants saw a screen with a start button and four response boxes. Stimuli consisted of colored Xs (red, blue, green, and yellow) that appeared inside one of the four response boxes. The start button was gray with the word "Start" in black text and was located in the bottom center of the screen. We refer to the other four boxes as B1, B2, B3, and B4 from left to right (see Fig 1). The boxes on the outside (B1 and B4) appeared halfway up the center of the screen. The boxes in the center (B2, B3) appeared at the top. All four of these boxes started off as white outlines with black centers. Each box had the following dimensions: width = 160 pixels (1/8 of the screen's width), height = 100 pixels (1/8 of the screen's height). The boxes were 6° along the diagonal in visual angle.

## Procedure

The trial structure is shown in Fig 1. Participants were told they would see colored Xs and that they would have to click on their target. Half the participants searched for a green X and half for a red X. Participants started the trial by clicking on the start button. After clicking, there was a 500 ms delay where the start box was removed (but the other four boxes remained). This was followed by a flash of color. The outline of one of the boxes changed color for 75 ms. This flash of color was the cue and it matched the target on 50% of trials (i.e., green flash when searching for a green X) and mismatched on 50% (i.e., red flash when searching for a green X). This screen was followed by a screen showing the four boxes with the white outlines again for 50 ms. At this point, the mouse was reset to the bottom center of the start screen to ensure that all trajectories started from the same origin point. Finally, four colored Xs appeared inside the boxes on the last screen. Participants moved their mouse to the box containing their target and clicked on it. Note, participants did not have to click the X itself; they could click anywhere inside the boxes containing the correct X. Participants had 2000 ms to make their response. The locations of the colored Xs were randomized and counterbalanced such that the target appeared at each location an equal number of times. Trials were marked correct if participants clicked on the box containing the appropriately colored X (either red or green) within the response deadline (2000 ms). Participants were explicitly warned to ignore the flash as this was intended to distract them and compromise their performance. Participants completed 32 practice trials followed by five blocks of 96 trials each. The location of the target and the cue were randomized and counterbalanced within each block, meaning that the cue occurred on the target location about 25% of the time. A blank screen with a fixation dot in the center appeared

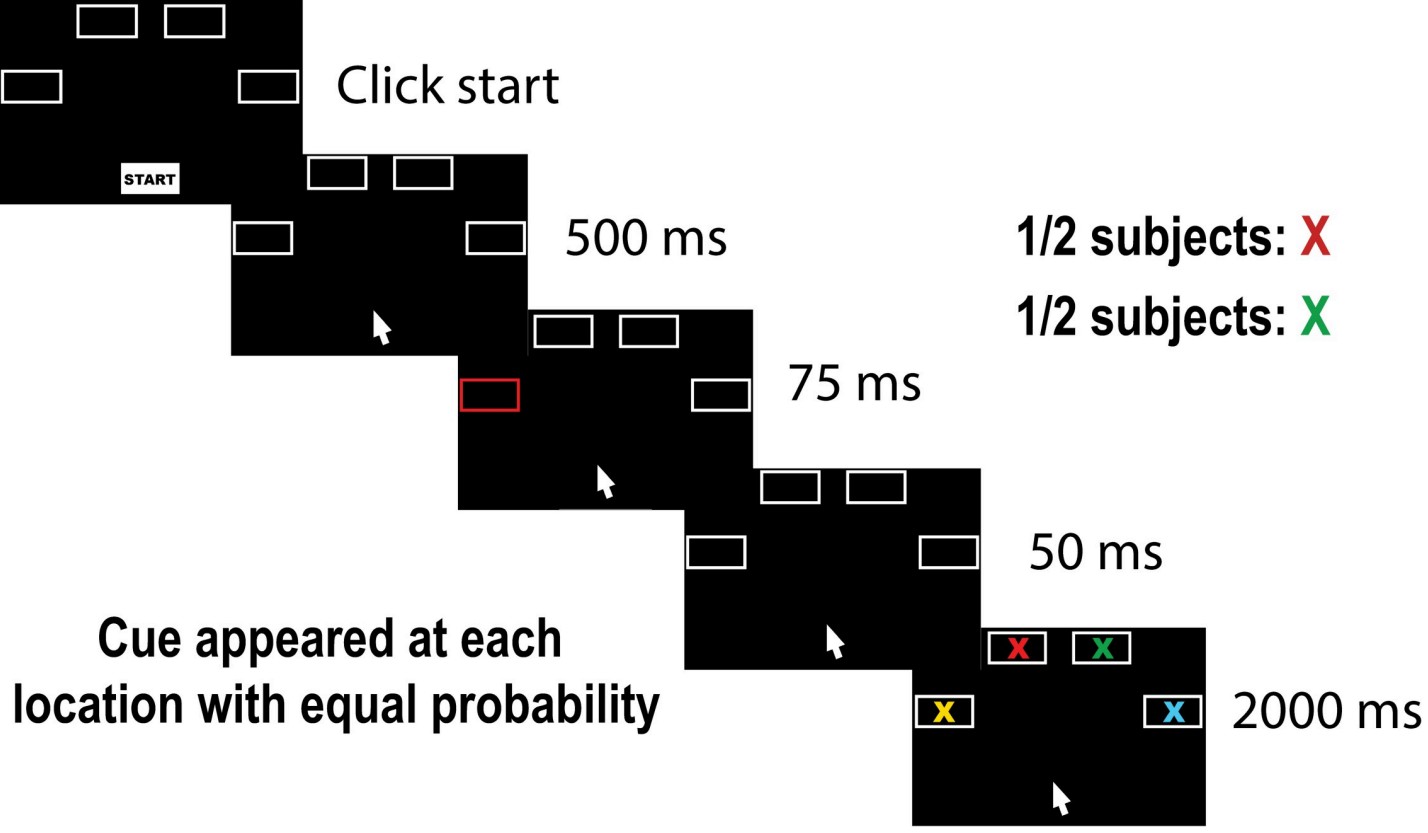

**Fig 1. Experiment 1 procedure.** Half of the participants searched for a green X and half searched for a red X. Participants clicked the start button to initiate the trial. Thereafter, there was a 500 ms delay. Then one of the four boxes was cued with a flash of color for 75 ms that either matched or mismatched the color of the target. The cue could appear in any of the four boxes (labeled B1-B4 from left to right). Finally, there was a 50 ms delay where the boxes were shown, followed by a screen with colored Xs inside the boxes. Participants clicked on the box containing the colored X they were searching for. Capture was evidenced by mouse trajectories that curved toward the cue when it appeared on the opposite side of the screen compared to the target.

for 1000 ms between trials. As a reminder, the boxes were labeled B1 to B4 from left to right. Trials where the target appeared at B1 or B4 were dropped from analyses as noted in the Data Exclusion section of our preregistration (consistent with previous research [35]). If the target was B2 and the cue appeared at B1 or B2, this was counted as a "same side" trial. However, if the target was B2 and the cue appeared at B3 or B4, this was counted as a "different side" trial. The converse is true for when the target appeared at B3.

After this first task, participants then completed the Self-Control Scale [13]. This scale has been found to be negatively associated with delayed discounting using preference based tasks [43], point based tasks [44], and, most importantly, using monetary based tasks (in the form of gift cards [45]). The scale consists of 36 statements (24 reverse scored), such as *I am good at resisting temptation* and *I am self-indulgent at times* (reverse scored). Participants rated the extent to which they agreed with each item on a scale of 1 (*not at all like me*) to 5 (*very much like me*). A single trait self-control score was created for each participant by calculating the reverse coded items and averaging across the 36 items *(M = 3.35, SD = .52, α = .90)*.

### Data preprocessing

Before conducting our analyses, we first excluded all practice trials, incorrect trials, and trials where the participant did not respond within the response deadline (2000 ms). In addition, we

filtered out trials where the response time was faster than 300 ms or slower than 3.5 standard deviations above the participant's mean as determined a priori (see the Data Exclusion section of the preregistration). Finally, only trials appearing at the top central boxes (B2 and B3) were analyzed. All data filtering and preprocessing were specified in the preregistration.

For ease of visualization and comparison, trajectories were remapped to the top left response so that all responses began in the same spot and ended at the same response box. Afterwards, trajectories were normalized to 101 steps [27, 29, 30] to ensure that responses had an equivalent number of coordinates regardless of differences in response time. This was done for visualization purposes. The dependent measures for each trajectory were calculated based on the raw trials (un-normalized). These dependent measures were aggregated up to the participant level for analyses.

## Results

### Manipulation check

In addition to our primary analyses reported below, we ran secondary analyses to investigate attention capture effects in general. These analyses can be seen as manipulation checks. Given that they are secondary to the primary purpose of the paper, we report these in the Supplementary Materials (https://osf.io/xnc7k/). Here, we merely note that these analyses are consistent with previous research of ours [35] and verify that our manipulation worked. Namely, capture occurred for both cues matching and mismatching the color of the target, as indicated by mouse trajectories that initially curved toward distractors when they occurred on the opposite side of the screen compared to the same side of the screen as the target, but capture was substantially smaller for cues not sharing the color of the target since trajectories corrected toward the target soon after this initial curvature. This demonstrates that both matching and mismatching cues captured attention but mismatching cues were easier to disengage from. Having established that captured did occur, we now turn our attention to the trait self-control analyses.

### Primary analyses

Our primary analyses used two mouse parameters to derive dependent variables: area under the curve and time of maximum deviation. Area under the curve (AUC) is calculated as the difference in area (measured in pixels) between a direct trajectory from the start position of the mouse to the target and the actual trajectory of the mouse movement; this measures the degree of spatial attraction to the competing, but ultimately unchosen, responses [46]. In contrast, time of maximum deviation (TMAD) is a temporal measure calculated as the latency in time between the beginning of the trial and the point of greatest divergence from a direct line to the target; this measure provides a temporal index of the resolution in attraction [47].

In order to test whether trait self-control affects attention capture, we created different sets of capture scores: one to primarily index attentional selectivity and one to primarily index disengagement (see Fig 2). We created a "selectivity" index by subtracting mouse parameters for trials featuring mismatching (non-target color) cues on the target-opposite side of the screen from trials featuring matching (target color) cues on the opposite side (i.e., Matching Cue Opposite Side–Mismatching Cue Opposite Side). This measure compares a person's distraction to goal-similar cues (e.g., red cue, red target) against their distraction to goal-dissimilar cues that mismatch the target (e.g., green cue, red target). For area under the curve, higher selectivity scores reflect greater distraction by goal-similar features (relative to goal-dissimilar features). For time of maximum deviation, higher scores reflect a longer period of distraction specifically for goal-similar relative to goal-dissimilar information.

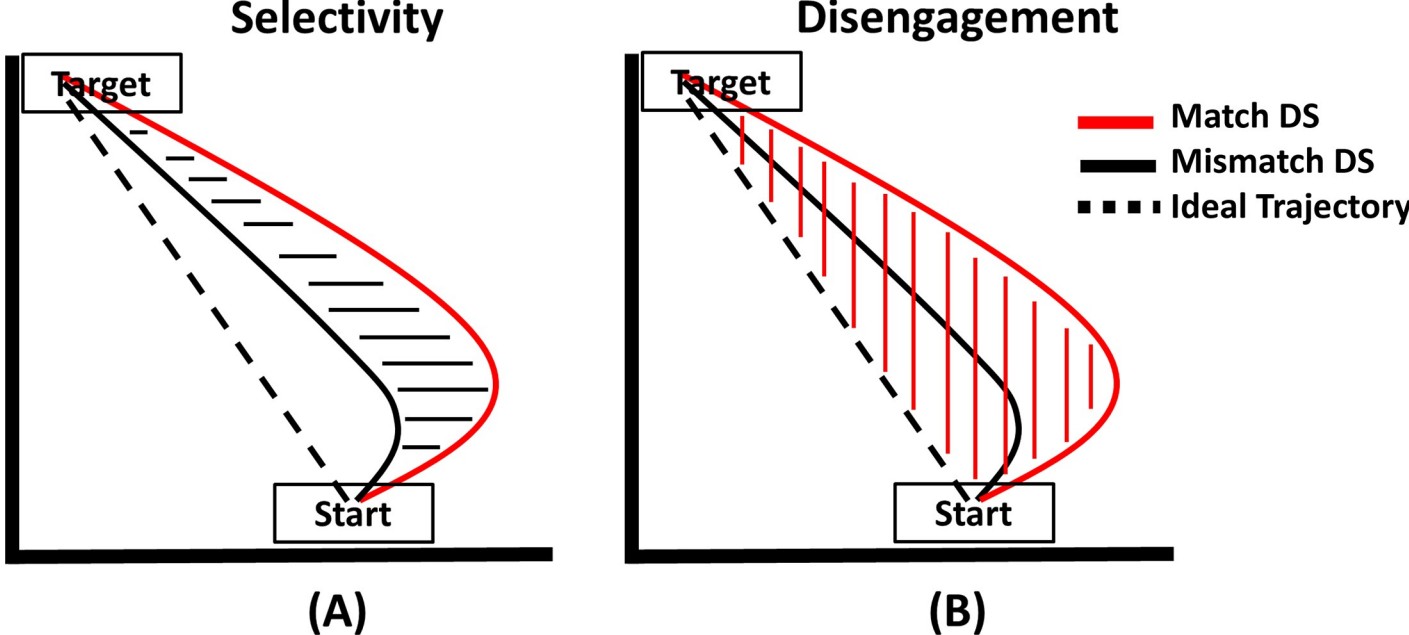

**Fig 2. Selectivity and disengagement.** Graphical representation of our selectivity and disengagement indices based on Area Under the Curve (AUC). (A) Selectivity is the difference in area under the curve between trials on which the distracting cue matched vs. mismatched the color of the target when the cue appeared on the opposite side of the screen as the target. (B) Disengagement is the area under the curve when the cue matches the color of the cue and the cue appears on the opposite side of the screen as the target. It corresponds to the difficulty in disengaging (i.e., inhibiting) the goal-similar cue.

In addition, our disengagement index was merely mouse parameters for trials on which the cue matched the color of the target and appeared on the side of the screen opposite the target. The greater the area under the curve for matching cues (e.g., red cue, red target) the more difficult it was for a person to disengage from the goal-similar distractor. Similarly, the greater the time of maximum deviation, the slower a person was overall from disengaging with the goal-similar distractor.

### Reliability analyses

Before any meaningful individual difference relationships can be explored, attention capture indices must first be examined for reliability. This is in response to previous research [48] which found that most attention capture paradigms produced capture scores with little to no reliability. We calculated reliability by splitting the data into odd and even trials, aggregating them to the participant level, calculating selectivity and disengagement indices, and then correlating the odd data set against the even data set. The reliability scores for our selectivity and disengagement indices are summarized in Table 1. Reliability for the disengagement measure

**Table 1. Primary analyses.**

| | Reliability Correlations | | | | | | TSC Correlations | | | | | |
|---|---|---|---|---|---|---|---|---|---|---|---|---|
| | AUC | | | TMAD | | | AUC | | | TMAD | | |
| | r | t-value | p | r | t-value | p | r | t-value | p | r | t-value | p |
| Selectivity | 0.67 | 8.94 | < .001 | 0.38 | 4.05 | < .001 | 0.02 | 0.233 | 0.816 | 0.06 | 0.546 | 0.587 |
| Disengagement | 0.85 | 15.65 | < .001 | 0.91 | 21.73 | < .001 | -0.01 | -0.094 | 0.925 | -0.04 | 0.364 | 0.717 |

Reliability and trait self-control correlations for selectivity and disengagement for area under the curve (AUC) and time of maximum deviation (TMAD).

was $r = .85$ for area under the curve derived index and $r = .91$ for time of maximum deviation derived index. Reliability for the selectivity index was $r = .67$ for area under the curve derived scores, but only $r = .38$ for time of maximum deviation derived scores. Overall, these reliability scores are either within the same range or much higher than similar measures derived from response times [48]. For their RT-based contingent capture paradigm, Roque, Wright, & Boot [48] report split-half reliability scores that ranged from as low as $r = .34$ (experiment 1) to as high as $r = .48$ (experiment 2). Thus, our reliability scores were comparable at worst, but considerably higher at best.

## Primary analyses

As shown in Table 1, trait self-control did not correlate with selectivity or disengagement for either area under the curve or time of maximum deviation. Bayes Factors were also calculated ($B_{10}$) to examine relative evidence for the null vs. alternative hypotheses. $B_{10}$ ranged from .13 to .15, indicating moderate support for the null [49]. For ease of visualization, we have provided a plot of the area under the curve broken down by participants with low, medium, and high self-control (see Fig 3). Trajectories look virtually identical across different levels of trait self-control.

Note, it is also possible that disengagement from goal dissimilar cues would be a better measure of an individual's trait self-control. To explore this, we reran the correlations using mouse tracking parameters for mismatching cues. This did not change the pattern of results (all $r$s < .1, all $p$s > .5). In addition, we also ran analyses controlling for other individual differences such as basic demographic information (age, gender, and parents' education and income). Controlling for these demographic variables did not change the pattern of results. See Table B in S2 Materials.

## Exploratory analyses

Given these null results, we also conducted exploratory correlations to test the relationships between trait self-control and initiation time and response time. Initiation time represents the time between a trial starting and the first movement of the mouse, whereas response time represents the time it takes to make one's selection. As shown in Table 2, neither the selectivity nor disengagement measures for response time and initiation time correlated with trait self-control scores. Note that reliability was quite low for response time selectivity ($r = .23$) and initiation time selectivity ($r = .16$).

## Discussion

Overall, attention capture was unrelated to differences in trait self-control. Trait self-control did not correspond to differences in measures of capture nor in subsequent disengagement. This suggests that the relatively micro-level phenomenon of attention capture and the relatively macro-level personality trait of self-control are orthogonal to one another.

### A closer look: Trait self-control

Is it surprising that trait self-control did not affect attention capture? On the one hand it is surprising because a number of studies *have* found differences in response dynamics due to differences in trait self-control. For instance, prior research [50] presented participants with healthy and unhealthy food items (e.g., apple and cake) and had them pick the healthy option while tracking their mouse movements. Compared to participants with lower trait self-control, participants with higher self-control showed less conflict in their decision-making processes (i.e.,

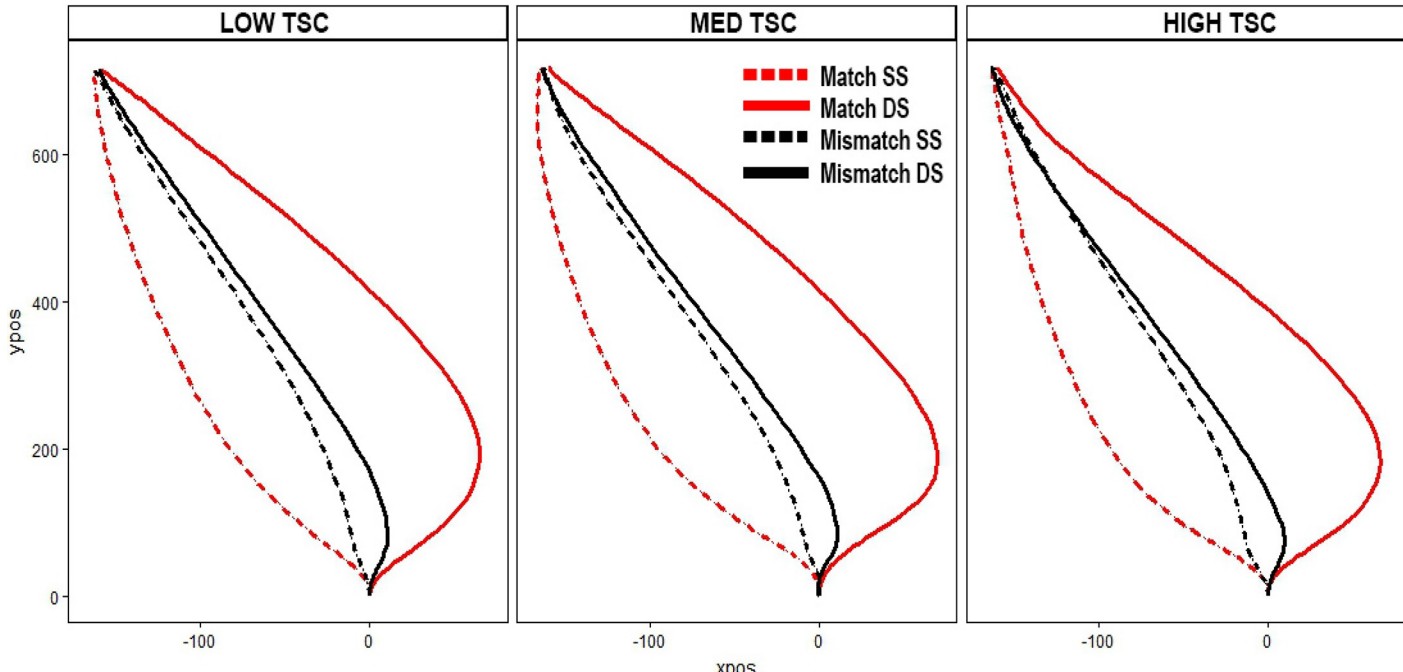

**Fig 3. Trajectories.** The trajectories across low, medium, and high trait self-control (TSC) groups. Note, data was analyzed continuously; these categorical groups are for visualization purposes only.

less area under the curve) and did so with smoother, less abrupt trajectories. In contrast, other research [51] also looked at trait self-control differences using mouse-tracking but found a different pattern of results. While they did not find spatial differences (e.g., area under the curve or maximum absolute deviation) between participants with high and low self-control, they did find temporal differences (i.e., time of maximum deviation) in trajectories. Participants with high self-control inhibited their responses quicker than low self-control participants, as evidenced by faster time of max deviation. These temporal differences are consistent with research showing that "tastiness" information influences trajectories sooner than "healthfulness" information for people with low self-control [52–53]. Nonetheless, our study found neither spatial nor temporal differences within the domain of attention capture, which is inconsistent with prior research linking trait self-control to spatial differences reflective of magnitude differences [50] and to temporal differences reflective of processing differences [51, 52]. Perhaps this is due to the relatively different cognitive processes involved in the experimental paradigms. The contingent capture paradigm is an abstract, relatively low-level task

**Table 2. Exploratory analyses.**

| | Reliability Correlations | | | | | | TSC Correlations | | | | | |
|---|---|---|---|---|---|---|---|---|---|---|---|---|
| | RT | | | IT | | | RT | | | IT | | |
| | r | t-value | p | r | t-value | p | r | t-value | p | r | t-value | p |
| Selectivity | 0.23 | 2.168 | 0.032 | 0.16 | 1.607 | 0.111 | 0.13 | 1.338 | 0.184 | -0.07 | -0.66 | 0.511 |
| Disengagement | 0.9 | 21.039 | < .001 | 0.95 | 31.208 | 0.001 | -0.06 | -0.567 | 0.572 | -0.04 | -0.445 | 0.657 |

Reliability and trait self-control correlations for selectivity and disengagement for response time (RT) and initiation time (IT).

whereas the selection of dietary preferences is a relatively high-level task that depends on motivation and values and closer approximates real world behavior.

On the other hand, it may not be surprising that self-control did not affect attention capture. There is evidence that successful self-regulation may largely depend on automatized, habitual processes rather than deliberative ones [15, 24– 25, 55]. One possibility is that self-control differences *do* affect attention capture but that self-control strategies must be deployed deliberately and repeatedly in order to later automatize selection and/or disengagement from attention capture. Prior work demonstrates that briefly training participants to automatically approach goal-supporting targets (i.e., pushing a joystick toward healthy stimuli) and avoid goal-conflicting targets (i.e., pulling a joystick away from unhealthy stimuli) led to increased efficiency in an attention task as well as to a higher likelihood of selecting a healthy, but not unhealthy, snack (similar results were demonstrated with school vs. party-related stimuli and intentions to study; [54]). Hence, future research may benefit by testing whether training participants in the attention capture task would lead to more efficient automatization of selectivity and disengagement for people higher, but not lower, in trait self-control. As a proxy for a training task, we evenly divided trials into early, middle, and late and then correlated selectivity and disengagement scores with trait self-control scores. Learning within the task did not affect correlations (all $p$s > .5).

The goal in the current study was to test whether the mechanisms related in trait self-control are related to the mechanisms involved in the relatively low-level phenomenon of attention capture. Although the mechanisms appear to be unrelated, it is quite possible that trait self-control would be correlated with "higher order" tasks, such as a mouse-tracking task where participants select between healthy and unhealthy items. Indeed, we have previously cited some research showing that this the case [50–52]. As such, future research may wish to investigate exactly how "low" the mechanisms of trait self-control operate. At what point in the continuum between perceptual tasks and cognitive tasks does trait self-control function?

## Reliability

Our results contribute to the attention capture literature in general by providing a paradigm that has a greater degree of reliability, particularly with spatial metrics such as area under the curve. Previous research [48] found that reliability across classic attention capture paradigms was generally low. In bottom-up paradigms, split-half reliability ranged from as low as .12 in the irrelevant singleton task to no higher than .28 in the onset cueing task; similarly, in top-down paradigms, split-half reliability ranged from as low as .34 in contingent cuing (experiment 1) to no higher than .48. In contrast, our modified contingent capture paradigm had higher reliability. The reliability of area under the curve derived measures—which provide crucial spatial information of capture—was .67 for our selectivity measure (a difference measure) and .85 for our disengagement measure. In addition, the time of maximum deviation had a reliability of .91 for our disengagement measure. These reliability scores suggest that continuously tracking a participant's mouse is more reliable a measure of capture than outcome based temporal measures like response time.

## Limitations

The current study has a number of limitations to consider. For one, although we created selectivity and disengagement scores as proxies for different mechanisms of attention capture, exploratory analyses found a number of correlations amongst the scores (e.g., AUC: selectivity and disengagement .864; a full correlation table is shown in Table C in S2 materials). These correlations could be indicative of several different things. One possibility is that selectivity

and disengagement scores are measuring the same mechanism, despite our intention to have them measure different mechanisms of attention capture. Another possibility is that selectivity and disengagement scores measure different mechanisms, but the two mechanisms are highly related to one another. For example, it may be that selectivity and disengagement are independent mechanisms, but people with high selectivity may tend to also have high disengagement. Future research is needed to tease these possibilities apart.

## Conclusion

Our study suggests that attention capture is impenetrable by trait self-control. Despite the mechanistic similarities, it appears self-control and visual attention recruit and rely on different mechanisms of information selection and behavioral inhibition. The ability to say no to a slice of cake and the ability to resist attention capture appear to be independent and orthogonal to one another. One may be distracted by a cake and not want to eat it too.

## Supporting information

**S1 Fig. Area under the curve.** Errors bars represent confidence intervals.
(EPS)

**S2 Fig. Response time.** Errors bars represent confidence intervals.
(EPS)

**S3 Fig. Initiation time.** Errors bars represent confidence intervals.
(EPS)

**S4 Fig. Time of maximum absolute deviation.** Errors bars represent confidence intervals.
(EPS)

**S1 Materials.**
(DOCX)

**S2 Materials.**
(DOCX)

## Author Contributions

**Conceptualization:** Michael A. Dieciuc, Heather M. Maranges, Walter R. Boot.

**Data curation:** Michael A. Dieciuc, Heather M. Maranges.

**Formal analysis:** Michael A. Dieciuc, Heather M. Maranges, Walter R. Boot.

**Investigation:** Michael A. Dieciuc, Heather M. Maranges, Walter R. Boot.

**Methodology:** Michael A. Dieciuc, Heather M. Maranges, Walter R. Boot.

**Writing – original draft:** Michael A. Dieciuc, Heather M. Maranges, Walter R. Boot.

**Writing – review & editing:** Heather M. Maranges, Walter R. Boot.

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
