## [Decision Letter · Decision Letter 0]

6 Sep 2019

PONE-D-19-20912

Trait Self-Control Does Not Predict Attentional Control: Evidence from a Novel Attention Capture Paradigm

PLOS ONE

Dear Mr. Dieciuc,

Thank you for submitting your manuscript to PLOS ONE. After careful consideration, we feel that it has merit but does not fully meet PLOS ONE’s publication criteria as it currently stands. Therefore, we invite you to submit a revised version of the manuscript that addresses the points raised during the review process.

Thank you for submitting your very nice paper to PLOS One.  The paper  is almost ready for publication, but before I accept it I would like you to perform the following additional analyses of the data:

1) As suggested by referee 1, please compare behavior  in the early rounds of a session with behavior in the later ones.

2) As suggested by referee 2, please run additional analyses that include the available demographic information. If you find that this has little impact and/or that the controls are statistically insignifcant then you could mention that in a footnote; otherwise it would seem natural to include it in the text.

3) As also suggested by referee 2, please do some analysis of the te the influence of self-control on attentional capture- regression is one way to do that but you may prefer others.

Both referees also make interesting suggestions for further experiments. I encourage you to consider following up on these suggestions in future work.

sincerely

Drew Fudenberg

We would appreciate receiving your revised manuscript by Oct 21 2019 11:59PM. To enhance the reproducibility of your results, we recommend that if applicable you deposit your laboratory protocols in protocols.io, where a protocol can be assigned its own identifier (DOI) such that it can be cited independently in the future. For instructions see: http://journals.plos.org/plosone/s/submission-guidelines#loc-laboratory-protocols

We look forward to receiving your revised manuscript.

Kind regards,

Drew Fudenberg

Academic Editor

PLOS ONE

Journal Requirements:

Additional Editor Comments (if provided):

Reviewers' comments:

Reviewer's Responses to Questions

**Comments to the Author**

1. Is the manuscript technically sound, and do the data support the conclusions?

Reviewer #1: Yes

Reviewer #2: Partly

2. Has the statistical analysis been performed appropriately and rigorously? 

Reviewer #1: Yes

Reviewer #2: No

3. Have the authors made all data underlying the findings in their manuscript fully available?

Reviewer #1: Yes

Reviewer #2: Yes

4. Is the manuscript presented in an intelligible fashion and written in standard English?

Reviewer #1: Yes

Reviewer #2: Yes

5. Review Comments to the Author

Reviewer #1: This study investigates whether trait self-control correlates with individual susceptibility to attention capture. Self-control is measured using self-reported perception of self-control with various questions on a scale, while susceptibility to attention capture is measured using a novel experimental paradigm which relies on mouse tracking.

The article makes 2 main contributions.

The first contribution is methodological: the authors propose a novel experimental task to measure low-level attention capture and show that their resulting (pre-registered) quantitative indexes are much more reliable than those previously used in the literature (as measured by the degree of internal correlation within-subject).

The second key contribution is a null result: trait self-control is not correlated with susceptibility to attention capture. This contribution is significant because:

- Both self-control and resistance to attention capture are thought to employ similar mechanisms, relying on inhibitory control, albeit at a high cognitive level for self-control and at a low cognitive level for attention capture;

- Self-control and resistance to attention capture have been shown to correlate in an experimental paradigm which involves utility-relevant distractions (e.g. food items). Yet it was unclear if this correlation would survive if the attention capture environment was made lower-level with purely perceptual stimuli. This study provides a negative answer to this question, suggesting the correlation does rely on the nature of the distraction and is not due to low-level cognitive mechanisms involved in attention control (note: this is an important consideration and I find the article would benefit if this was mentioned in the introduction, not only in the conclusion) ;

- Both self-control and susceptibility to attention capture have been shown to correlate with working memory.

Minor comments:

- An eye-tracking based version of the experiment would seem like a more natural paradigm, because it would not rely on the assumption that “the partial and tentative conclusions of perceptuo-cognitive systems continuously feed into and affect the motor system” (which the authors correctly mention). The article would benefit from a discussion justifying the choice of mouse-tracking, unless it was a purely practical choice.

- In the “Predictions” section, the verb “affect” is repeatedly used (“self-control may affect only selectivity” etc.), suggesting a causal link. However nothing in the literature cited nor in the data collected supports a causal relationship. Hence it would be more appropriate to use a verb such as “correlates with”.

- The self-control scale used suffers from being purely based on a self-report. The study would benefit from comments about how well the specific scale used here predicts revealed preferences measures such as actual intertemporal choices.

- The authors mention that it would be interesting to investigate whether learning arises and self-control does help resist attention capture but only after some training. I agree that this is an interesting follow-up question. But while the authors have no data on training, they do have a lot of trials per participant. They thus could provide a partial answer by comparing behavior in the later trials to behavior in the earlier trials in the experiment. It would be beneficial to run such tests and add some words on these.

Reviewer #2: Summary:

This work measures attention capture—the tendency to briefly respond to extraneous stimuli before returning to a primary task—via a mouse-tracking experiment, with the goal of assessing the correlation between that behavior and a questionnaire measuring self-control. Specifically, the work seeks to examine the effect of self-control on selectivity (whether the strength of attention capture increases as the distractor becomes more similar to the goal) and disengagement (how easy it is for the subject to return to the goal after being distracted). In the experiment, subjects are tasked with clicking a colored X which appears in one of four boxes, and are distracted by the flashing outline of a randomly-selected box before doing so. By changing the color of the distracting stimulus to match or differ from that of the goal, the work replicates findings in the literature wherein similar stimuli are more effective in capturing attention.

To provide a description of the magnitude of attention capture, the work computes both the distance (in pixels) added to a straight-line trajectory by the distracting stimulus and the time taken by the subjects to begin moving their mouse towards the goal. Selectivity is measured by comparing each subject’s behavior when the distractor’s color matched the goal to their behavior when it did not; disengagement is measured by restricting analysis to the case where the distractor’s color matched the goal, but the distractor was on the opposite side of the screen. The work splits the data set in two and examines between-subsample correlations in both the attention-capture and self-control measures, finding a high degree of reliability. However, the work finds near-zero correlations between self-control and their attention capture measures, and the mouse trajectories of the participants show no clear differences regardless of self-control scores. This result is taken as grounds to claim that attention capture and self-control operate through different cognitive mechanisms.

Comments:

Following the seven PLOS ONE Criteria for Publication,

(1) Primary results of original research: criterion met; no further comments.

(2) Results not published elsewhere: criterion met, this work has only been submitted to PLOS ONE; no further comments.

(3) Experiments, statistics, analyses are high-standard and described in detail: descriptions are clear and reproducible, but some additional analysis is needed to confirm the reliability and consistency of the measures of attention capture and address other potential variables of interest

a. It would be good to see the correlation between the two measures of attention capture, both within-sample (i.e., compare each participant to themselves) and “out-of-sample” (i.e., compare on aggregate or using subsampling). These results would provide stronger evidence of whether these measures are really capturing the same behavior.

b. Along the lines of (2), there are no demographic/socioeconomic/other controls included in the correlation analysis. The work notes that attention-capture and self-control may vary due to individual-level traits (e.g. video-game-playing); is the goal of the study to capture the correlation after taking into account these personal idiosyncrasies or to extract some more “baseline” measure, or one that can be adjusted for interactions with various personal factors? If it is the latter, then including controls would help to provide a clearer picture of how self-control and attention capture interact without the influence of particular personal traits.

(4) Conclusions supported by the data: some scope for clearer causal analysis and further experiments to more directly involve the processes of self-control

a. In addition to examining the correlation between attention capture and self-control, it seems valuable to make some effort at examining the influence of one on the other, e.g., regressing attention-capture measures on self-control scores. While there is potential for reverse causality (sensitivity to attention capture influences the ability to develop self-control), given that self-control is the macro-level process it seems more likely to be the overall driver of behavior. Especially if both attention-capture measures are used, care must be taken in interpreting the coefficients, but this approach could still provide another angle on the relationship between the two variables.

b. The measure of disengagement includes only goal-similar distractors; it is unclear why this is the optimal measure of disengagement. It may be that self-control does not influence goal-similar attention capture but does improve disengagement from goal-dissimilar attention capture—in the dessert vs. healthy eating example, self-control may improve the ability to reject a goal-dissimilar slice of cake, but not a goal-similar fruit tart. Investigating the effect of self-control on goal-dissimilar distractors, or the overall measures of attention capture not split by goal-similarity, would provide a richer picture of this effect.

c. In addition to using questionnaire measures of self-control, and in order to more directly invoke self-control behavior, additional tasks that take advantage of the mouse-tracking framework could be implemented. The work notes that self-control can vary according to “situational demands,” so using tasks that more directly invoke self-control (e.g., clicking on the “healthy” food from among various “unhealthy” options) would provide a valuable point of comparison to the more abstract task used here.

(5) Presented in intelligible fashion: criterion met; no further comments.

(6) Meets ethics standards: criterion met; IRB documentation provided, no further comments.

(7) Data availability: criterion met; analysis is preregistered and data is available, no further comments.

Conclusions

This work lays out an experimental method that, while it could use refinement, provides a great level of detail about the temporal and spatial behaviors associated with attention capture without requiring particularly resource-intensive hardware and software. There is room to take further advantage of this method to obtain a more complete picture of attention capture, which may include a greater variety of tasks, the introduction of more true-to-life goals and distractions, and clearer measures of disengagement in different contexts. In addition to these extensions of the existing experiment—extensions that could provide more direct experimental evidence of the role of self-control—the analysis of existing data could be strengthened by considering alternative statistical approaches and a richer set of potential independent and dependent variables. These modifications would help take advantage of what appears to be a powerful approach to answering important psychological questions.

6. PLOS authors have the option to publish the peer review history of their article (what does this mean?). If published, this will include your full peer review and any attached files.

Reviewer #1: No

Reviewer #2: No

---

## [Author Response · Author response to Decision Letter 0]

22 Oct 2019

We would like to thank the reviewers for their overall positive assessment of the manuscript. We found their feedback to be helpful in clarifying some ambiguities and in strengthening the message of the manuscript. Below we have distilled each reviewers major suggestions and provided a detailed explanation of how they were addressed. 

 

Reviewer # 1

-I find the article would benefit if this [the lack of correlation between attention capture and TSC] was mentioned in the introduction, not only in the conclusion)

Thank you for this suggestion, we now preview this result in the introduction on page 10.

“To preview our results, our data are most consistent with the fourth possibility. Namely, self-control scores did not correlate with either selectivity or disengagement. This suggests that attention capture and trait self-control operate via different mechanisms.” 

-The article would benefit from a discussion justifying the choice of mouse-tracking [compared to eye-tracking], unless it was a purely practical choice.

 We addressed this by adding the following text to the manuscript, pages 8-9: 

“Our decision to use mouse-tracking instead of a similar methodology like eye-tracking was based on a number of advantages related to measuring how cognitive processes play out over time. While saccades are largely ballistic in nature and their flight paths are often completed in tens of milliseconds, mouse-tracking unfolds over a much larger timescale, providing a rich look at both spatial and temporal dimensions of behavior. On another level, mouse-tracking was chosen over eye-tracking for a number of practical reasons. Mouse-tracking software is freely available (Kieslich & Henninger, 2017; Freeman & Ambady, 2010) to anyone who has access to a computer. This high-accessibility makes it an appealing method of conducting research, one that is particularly conducive to encouraging replication and reproducibility. Finally, mouse-tracking is a relatively new methodology compared to eye-tracking, especially within the field of attention capture and self-control; thus, given the novelty of the methodology, we wanted to deepen the field’s knowledge.” 

In the “Predictions” section, the verb “affect” is repeatedly used…suggesting a causal link. However nothing in the literature cited nor in the data collected supports a causal relationship. Hence it would be more appropriate to use a verb such as “correlates with”.

Thank you for pointing out this mistake. The text has been changed so that affects has been replaced with “correlates with.”

The self-control scale used suffers from being purely based on a self-report. The study would benefit from comments about how well the specific scale used here predicts revealed preferences measures such as actual intertemporal choices.

The trait self-control scale used here (Tangney, Baumeister, & Boone, 2004) reflects a macrolevel disposition that effects a wide variety of behavioral outcomes, including those in the domains of school and work, eating and weight, interpersonal functioning, and wellbeing and adjustment (for meta-analysis, see de Ridder et al., 2012). With respect to future discounting in particular, self-control as assessed with this scale has been negatively associated with delayed discounting (e.g., Franco-Watkins, Mattson, & Jackson, 2016; Kool, McGuire, Wang, & Botvinick, 2013; Pang, Otto, & Worthy, 2015)

Franco‐Watkins, A. M., Mattson, R. E., & Jackson, M. D. (2016). Now or later? Attentional processing and intertemporal choice. Journal of Behavioral Decision Making, 29(2-3), 206-217.

Kool, W., McGuire, J. T., Wang, G. J., & Botvinick, M. M. (2013). Neural and behavioral evidence for an intrinsic cost of self-control. PloS one, 8(8), e72626.

Pang, B., Otto, A. R., & Worthy, D. A. (2015). Self‐Control Moderates Decision‐Making Behavior When Minimizing Losses versus Maximizing Gains. Journal of Behavioral Decision Making, 28(2), 176-187.

-The authors mention that it would be interesting to investigate whether learning arises and self-control does help resist attention capture but only after some training…They thus could provide a partial answer by comparing behavior in the later trials to behavior in the earlier trials in the experiment. It would be beneficial to run such tests and add some words on these.

We followed this suggestion and evenly split up our data into early, middle, and late trials. Thereafter, we created selectivity and disengagement measures and correlated them with trait self-control scores. We did this once for early trials and once for late trials. It seems that learning within the context of the experiment did not affect correlations (all ps > .05). The exact statistics for each of these correlations is shown in the table below. 

We explained this by adding the following text to a footnote in the manuscript on page 22: 

“As a proxy for a training task, we evenly divided trials into early, middle, and late and then correlated selectivity and disengagement scores with trait self-control scores. Learning within the task did not affect correlations (all ps > .5).” 

 

REVIEWER # 2

It would be good to see the correlation between the two measures of attention capture, both within-sample (i.e., compare each participant to themselves) and “out-of-sample” (i.e., compare on aggregate or using subsampling). These results would provide stronger evidence of whether these measures are really capturing the same behavior.

As requested we have provided a within-sample correlation matrix of the attention capture measures. We were not entirely certain what the reviewer meant by “out-of-sample” correlations. If the reviewer wishes to specify, we would be happy to follow up with it. 

The correlation matrix below shows a number of correlations amongst our measures (e.g., AUC: selectivity and disengagement, r = .864). While these correlations do, in fact, suggest that the selectivity and disengagement scores are measuring the same overall behavior, it does not tease apart whether selectivity and disengagement scores are measuring the same or different mechanisms. For instance, the high correlation may suggest that selectivity and disengagement scores are actually measuring the same single mechanism of attention capture. However, another possibility is that selectivity and disengagement are measuring two different mechanisms that are themselves highly rated. For instance, people with high selectivity may tend to also have high disengagement. The current study cannot tease these different possibilities apart. To address this, we added the correlation table to our supplementary materials (Table S3) and the following text to the manuscript on p23: 

“The current study has a number of limitations to consider. For one, although we created selectivity and disengagement scores as proxies for different mechanisms of attention capture, exploratory analyses found a number of correlations amongst the scores (e.g., AUC: selectivity and disengagement .864; a full correlation table tis shown in Table S3). These correlations could be indicative of several different things. One possibility is that selectivity and disengagement scores are measuring the same mechanism, despite our intention to have them measure different mechanisms of attention capture. Another possibility is that selectivity and disengagement scores measure different mechanisms, but the two mechanisms are highly related to one another. For example, it may be that selectivity and disengagement are independent mechanisms, but people with high selectivity may tend to also have high disengagement. Future research is needed to tease these possibilities apart.”

Along the lines of (2), there are no demographic/socioeconomic/other controls included in the correlation analysis. The work notes that attention-capture and self-control may vary due to individual-level traits (e.g. video-game-playing); is the goal of the study to capture the correlation after taking into account these personal idiosyncrasies or to extract some more “baseline” measure, or one that can be adjusted for interactions with various personal factors? If it is the latter, then including controls would help to provide a clearer picture of how self-control and attention capture interact without the influence of particular personal traits.

This is a great point. Although we did not collect data on individual differences such as video game playing (which we mention as relevant to highlight that individual differences can modulate attention capture and disengagement processes), we collected data on basic demographic variables, such as age, gender, and parents’ education and income. (Note that because our sample includes student participants, their educational attainment and income do not really vary.) Our primary concern is the association between trait self-control and individual differences in distractibility, namely, attention capture and disengagement. Hence, although we are not focused on the interaction between self-control and demographic variables here, it is important to control for them.

Accordingly, we conducted correlation analyses controlling for these variables. Notably, this did not change the pattern of results that emerged without controls. This table is included in the Supplementary Materials and the results are mentioned in footnote 2 of the manuscript. 

 Pearson Correlations, controlling for age, gender, ethnicity, race, parental education, and parental income. 

 TSC AUC_SELECTIVITY AUC_DISENGAGE TMAD_SELECTIVITY TMAD_DISENGAGE

TSC — 

AUC_SELECTIVITY .057 — 

AUC_DISENGAGE .026 .866*** — 

TMAD_SELECTIVITY .121 -.110 .043 — 

TMAD_DISENGAGE -.036 -.643*** -.684*** .004 —

 *p < .05, **p < .01, ***p < .001

 

It seems valuable to make some effort at examining the influence of one on the other, e.g., regressing attention-capture measures on self-control scores.

We ran linear regressions and used our attention capture scores to predict trait self-control. As shown in the table below, neither attention capture score significantly predicted trait self-control scores. 

 B SE B β t p

 Intercept 3.362 1.03E-01 1.03E-16 32.745 < .001

AUC Selectivity 2.58E-06 4.13E-06 0.125 0.623 0.535

 Disengagement -2.00E-06 3.42E-06 -0.118 -0.586 0.559

 Intercept 3.49 0.344 3.487 10.147 < .001

TMAD Selectivity 0.0012956 0.002 0.056 0.551 0.583

 Disengagement 0.0003971 0.001 -0.038 0.373 0.71

The measure of disengagement includes only goal-similar distractors… Investigating the effect of self-control on goal-dissimilar distractors, or the overall measures of attention capture not split by goal-similarity, would provide a richer picture of this effect.

To address the possibility that goal-dissimilar distractors might be a better measure of disengagement, we ran correlations between trait self-control and area under the curve and time of maximum deviation on mismatching cues. As shown in the table below, none of the measures correlated. 

 r t-value p

AUC -0.06 -0.57 .570

TMAD -0.06 -0.561 .576

We supplemented the text with the following footnote found on page 17: 

“Note, it is also possible that disengagement from goal dissimilar cues would be a better measure of an individual’s trait self-control. To explore this, we reran the correlations using mouse tracking parameters for mismatching cues. This did not change the pattern of results (all rs < .1, all ps > .5).”

 

In addition to using questionnaire measures of self-control, and in order to more directly invoke self-control behavior, additional tasks that take advantage of the mouse-tracking framework could be implemented. The work notes that self-control can vary according to “situational demands,” so using tasks that more directly invoke self-control (e.g., clicking on the “healthy” food from among various “unhealthy” options) would provide a valuable point of comparison to the more abstract task used here.

We agree with the reviewer that these are interesting questions to ask and worthy of further research. Throughout the paper we cited a few studies that have examined trait self-control’s relationship to nutritional decisions in a mouse-tracking framework. We more explicitly address this by adding in the following text to page 22: 

“The goal in the current study was to test whether the mechanisms related in trait self-control are related to the mechanisms involved in the relatively low-level phenomenon of attention capture. Although the mechanisms appear to be unrelated, it is quite possible that trait self-control would be correlated with “higher order” tasks, such as a mouse-tracking task where participants select between healthy and unhealthy items. Indeed, we have previously cited some research showing that this the case (Stillman, Medvedev, & Ferguson, 2017; Gillebaart et al., 2016; Sullivan et al., 2015). As such, future research may wish to investigate exactly how “low” the mechanisms of trait self-control operate. At what point in the continuum between perceptual tasks and cognitive tasks does trait self-control function?”

---

## [Editor Report · Decision Letter 1]

24 Oct 2019

Trait Self-Control Does Not Predict Attentional Control: Evidence from a Novel Attention Capture Paradigm

PONE-D-19-20912R1

Dear Dr. Dieciuc,

Thank you for your responsive revision,  I am now happy to accept the paper for publication. Bolierplate follows below

----.

With kind regards,

Drew Fudenberg

Academic Editor

PLOS ONE
---

## [Editor Report · Acceptance letter]

21 Nov 2019

PONE-D-19-20912R1 

Trait Self-Control Does Not Predict Attentional Control: Evidence from a Novel Attention Capture Paradigm 

Dear Dr. Dieciuc:

I am pleased to inform you that your manuscript has been deemed suitable for publication in PLOS ONE. Congratulations! Your manuscript is now with our production department. 

With kind regards,

on behalf of

Dr. Drew Fudenberg 

Academic Editor

PLOS ONE